# U3D: Unlocking the Video Prior for High Fidelity Sparse Novel View Synthesis and 3D Generation

## Abstract

Trained on massive datasets, video diffusion models have shown strong generative priors for novel view synthesis tasks. Existing methods finetune these models to synthesize 360-degree orbit videos from input images. While these methods demonstrate the pretrained models' generalization ability, they are limited by the assumption of temporal attention and struggle to generate highly consistent results. Additionally, generating novel views as a sequence of twenty or more frames incurs high computational costs compared to sparse view synthesis methods. Sparse novel view synthesis methods finetuned from traditional 2D diffusion models, on the other hand, can generate highly consistent images from arbitrary camera positions but suffer from poor generalization, leading to unsatisfactory results on out-of-domain inputs. In this paper, we explore leveraging video diffusion models' rich generative priors to enhance sparse novel view generation models. Specifically, we investigate the generation process of video diffusion models and unearth key observations to extract geometrical priors from them. Based on this, we propose a novel framework, U3D, for sparse novel view synthesis. U3D includes a geometrical reference network to integrate these priors into the sparse novel view synthesis network and a temporal enhanced sparse view generation network to preserve pretrained temporal knowledge. By leveraging the significant generative priors from video diffusion models, our framework can synthesize highly consistent sparse novel views with strong generalization ability, which can be reconstructed into high-quality 3D assets using feed-forward sparse view reconstruction methods.

## 1 Introduction

The explosion of diffusion models has unlocked new paradigms for various downstream tasks, especially novel view synthesis. Existing methods, such as SV3D Voleti et al. (2024), finetune off-the-shelf video diffusion models Blattmann et al. (2023) on 3D rendered datasets to generate orbit videos from input images. While these methods largely preserve the generative priors from pretrained video diffusion models and yield reasonable performance, they are limited by the strong assumption of temporal attention and struggle to generate highly consistent 3D images with large camera movements. Additionally, video diffusion models generate sequences of 20 or more frames, leading to higher computational costs and slower generation speeds compared to sparse view synthesis methods.

In contrast, sparse novel view generation methods Shi et al. (2023b); Long et al. (2024) generate a small number of (one to six) novel views with arbitrary camera positions. These methods first generate consistent novel-view images and then use sparse view reconstruction models to reconstruct the generated 3D assets. The main advantages of these methods are: i) Computational efficiency: involving fewer target views, these methods have lower computational costs and faster inference speeds. ii) Higher 3D consistency: compared to video diffusion models, sparse view generation methods use 3D attention to learn correspondences across the entire synthesized views, maintaining 3D consistency with large camera movements. iii) Better generation flexibility: with dense 3D attention across the entire image, sparse view generation methods can generate views with arbitrary camera positions, without the constraint of sequence continuity in video diffusion models.

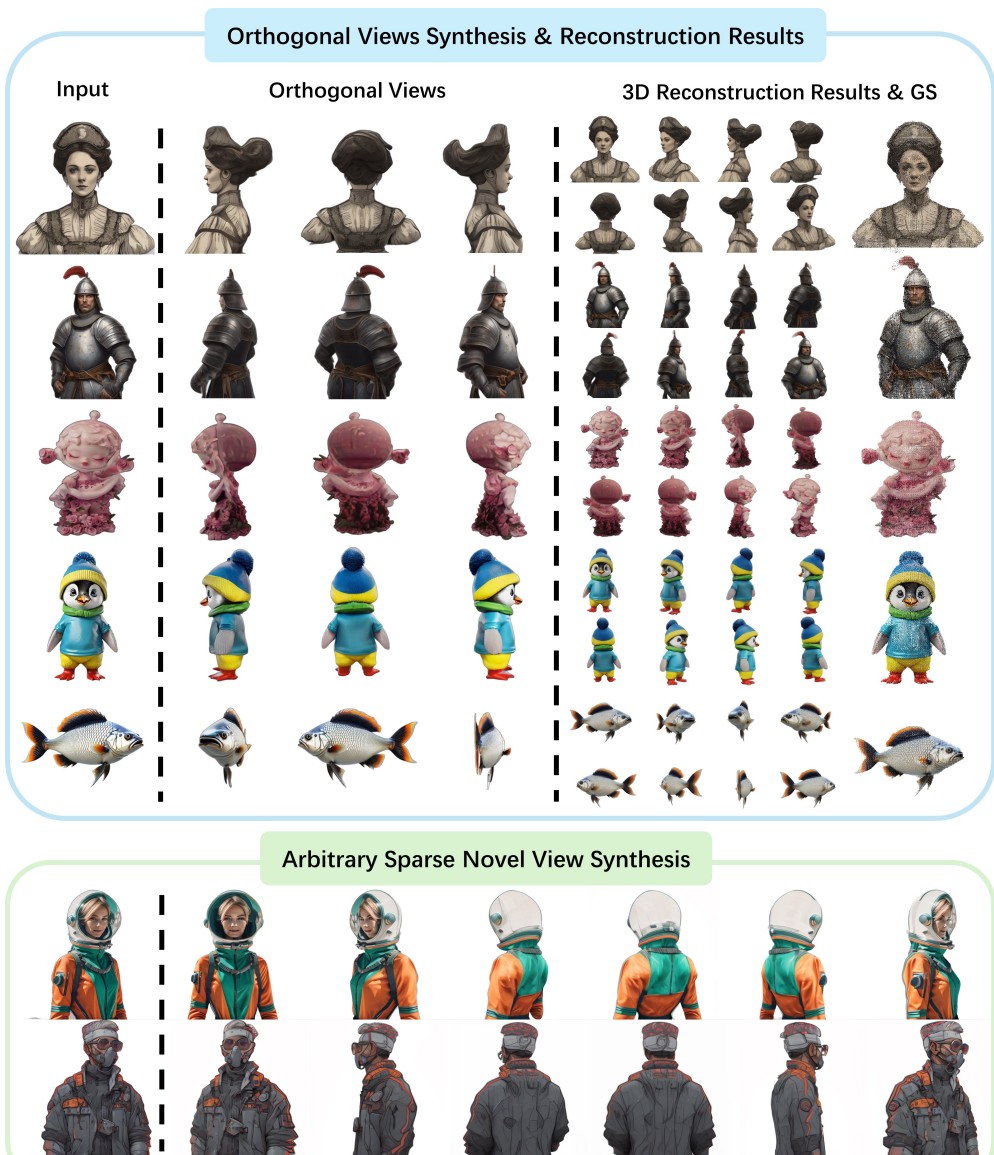

Figure 1: Give an input image and target camera positions, U3D is capable of synthesizing $512 \times 512$ high-quality sparse novel view images. We show the orthogonal views synthesis results here together with the reconstruction results from the Gaussian Reconstruction Model. Our model is capable of synthesizing arbitrary views of the input images as shown in the bottom of the figure.

However, most existing sparse view generation methods Wang & Shi (2023); Shi et al. (2023b); Long et al. (2024); Li et al. (2024) are finetuned from 2D diffusion models Rombach et al. (2022), which lack novel view knowledge of an image. As a result, these methods have poor generalization abilities and struggle to generate satisfying results for out-of-domain inputs, limiting their application in real-world scenarios. Therefore, we raise the question: Can we unlock generative priors in video diffusion models to enhance the generation quality and stability of sparse novel view synthesis methods?

To address this problem, we present U3D, a sparse novel view synthesis framework that unlocks the generative prior from video diffusion models for high-fidelity novel view generation. Specifically, we conduct in-depth investigations into the generation process of video diffusion models and discover that the temporal features in the decoder block of the video diffusion U-Net provide rich ge-

ometrical priors for novel view synthesis with noisy images as inputs. This observation inspires us to use the video diffusion model directly as a geometrical reference network to enhance the generation quality of sparse novel view networks.

To integrate the prior features from the video diffusion model into the sparse view generation network, we introduce a simple lightweight module called the residual temporal adapter. The residual temporal adapter serves as a plug-in temporal attention layer to calculate correspondences between the generated novel views and the extracted video temporal feature priors. The output values are then added back to the original features as a temporal residual to guide the generation process. This enhances the sparse view generation process with dense geometrical video priors in the temporal dimension, leading to stronger generalization ability and synthesis stability. Moreover, we introduce an adaptive control module to dynamically modulate the control strength from the video priors, enabling the model to synthesize accurate results with priors extracted from noisy inputs of different scales. During training, only the parameters of the newly added temporal residual layers are trained, while the pretrained video diffusion model and the sparse view synthesis model remain frozen. Such paradigm is efficient and preserves the original sparse novel view synthesis networks' ability and accuracy.

Additionally, we investigate the roles of different temporal attention layers in the U-Net of video diffusion models. We find that temporal attention layers in the deeper blocks capture global information, benefiting the generation process even with large camera movements. Compared with existing methods that mainly finetune a sparse view generation model from a pretrained 2D diffusion model, we introduce a new baseline sparse novel view synthesis network, named the temporal enhanced sparse view synthesis network, by finetuning a sparse view generation model from a pretrained video diffusion model. Specifically, we extend the 2D attention layer in the original video diffusion model into a 3D attention layer by concatenating keys and values from different views and finetune the video diffusion model in a sparse novel view synthesis setting, preserving the original temporal structure to maintain global temporal knowledge in the deeper blocks. This allows the network to benefit from the temporal knowledge initialized from the pretrained video model and generate more realistic images. To further enhance the view conditioning ability of the proposed sparse view generation network, we introduce a camera-aware frame embedding to dynamically adjust the temporal embedding with different camera conditions.

The aforementioned methods are unified into a novel sparse novel view synthesis framework named U3D, capable of synthesizing high-quality $512 \times 512$ novel view images with arbitrary camera positions. Compared to video diffusion models such as SV3D Voleti et al. (2024), U3D exhibits better 3D consistency in the generated results while involving fewer frames which accelerates the overall generation process. We conduct qualitative and quantitative experiments on different datasets and demonstrate that U3D, benefiting from the strong generalization ability and geometrical stability provided by the video priors, achieves state-of-the-art performance compared to existing methods and generates high-quality, 3D-consistent novel view images.

## 2 RELATED WORK

### 2.1 3D GENERATION.

3D generation has been well-explored with different 3D representations including meshes Gao et al. (2022), voxels Zhou et al. (2021); Chan et al. (2021), point clouds Yang et al. (2019), SDF Or-El et al. (2022); Park et al. (2019); Cheng et al. (2023), Triplane Chan et al. (2022); Gupta et al. (2023). Traditional methods Jun & Nichol (2023); Nichol et al. (2022) predominantly trained on limited-scale 3D datasets, often fall short in generating intricate geometric structures with substantial diversity. The explosion of diffusion models has unlocked new paradigms for 3D generation tasks. Many methods have been proposed to distill 3D information from the pretrained large diffusion models, which have been demonstrated to provide sufficient generative priors learned from the massive training datasets. Specifically, Score Distillation Sampling (SDS) based methods Poole et al. (2022); Wang et al. (2024); Qian et al. (2023); Lin et al. (2023) formulate the generation as an optimization process and utilize 2D pretrained diffusion model to provide supervision on the unseen views of the target object to distill the 3D information from the 2D diffusion models. Although being able to generate realistic results, these methods suffer from slow convergence and janus problem caused by the lack of 3D understanding and camera control ability in the pretrained 2d diffusion

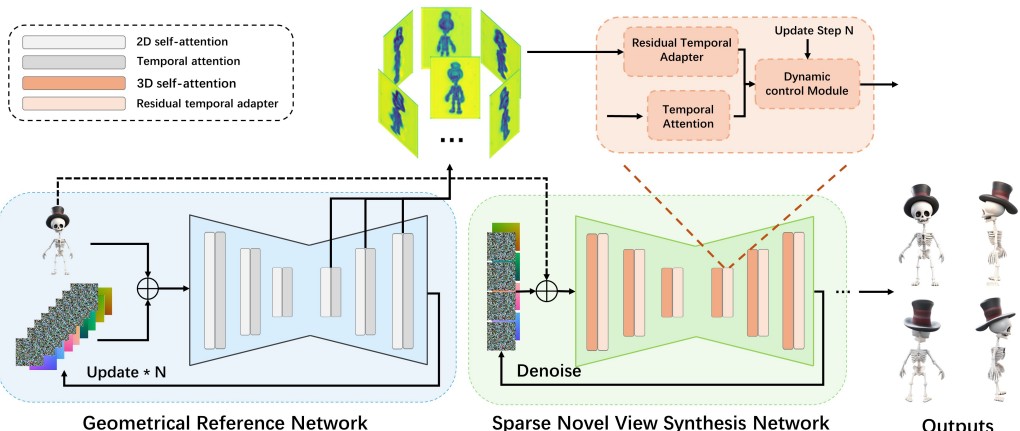

Figure 2: The overall framework of U3D. We adopt a pretrained video diffusion model as a geometrical reference network and extract the geometrical priors from the video diffusion model with a small number of denoise steps ($N = 8$) in our experiments. The extracted geometrical priors are then integrated into the proposed temporal enhanced sparse novel view synthesis network with the proposed Temporal Residual Adapter.

models. Another promising paradigm is to first generate multi-view images and then reconstruct the 3D shapes with NeRF, Gaussian Splatting or feed-forward large reconstruction networks Xu et al. (2024a;b); Li et al. (2023); Tang et al. (2024); Wei et al. (2024). Although achieving promising results, these methods still suffer from the local inconsistencies and the limited resolution of the input multi-view images and fail to generated 3D objects with complicated geometry and realistic textures.

## 2.2 NOVEL VIEW SYNTHESIS.

The success of diffusion models has opened a new door for the task of novel view synthesis. Zero123 Liu et al. (2023); Shi et al. (2023a) finetune the pretrained 2D diffusion model under different camera conditions to achieve arbitrary view conditioned generation. Sparse novel view synthesis mdethods like MVDream Shi et al. (2023b), for first time extend the original 2D self attention by concatenating keys and values in several views to achieve generation with 3D consistent multi-view images. Wonder3D Long et al. (2024) finetune the 2D diffusion model with cross-domain rgb-normal attention layers to facilitate the learning of geometry information of 2D diffusion models and enhance the 3D consistency of the generated outputs. However, constrained by poor generalization ability of 2D diffusion models, all of these methods struggle to generate satisfying results given out-of-domain inputs with complex geometry or textures. On the other hand, video diffusion models Blattmann et al. (2023) have been demonstrated to be able of providing strong generative priors for novel view synthesis tasks Xie et al. (2024); Zuo et al. (2024); Chen et al. (2024). SV3D Voleti et al. (2024) for the first time finetune a pretrained video diffusion model on the 3D rendered datasets to synthesize orbit 360 degree videos. Although yielding promising performance with great generalization ability, these methods are still limited by the strong assumption of temporal attention and fail to generate highly consistent novel view images with large camera movement.

## 3 METHODS

Given an image and arbitrary target camera positions as input, our goal is to synthesize 3D consistent novel view images that can be used to reconstruct 3D objects. To achieve this, we explore the possibility of adopting generative priors from video diffusion models to enhance the generation quality and generalization ability of sparse novel view synthesis networks. Specifically, we conduct in-depth investigations into the generation process of video diffusion models and propose a novel

geometrical reference network (Section 3.1) and a new sparse novel view synthesis network named the temporal enhanced sparse novel view synthesis network (Section 3.2).

## 3.1 GEOMETRICAL REFERENCE NETWORK

Video diffusion models have been demonstrated to provide generative priors and serve as strong initialization models for finetuning novel view synthesis models. Existing methods such as SV3D Voleti et al. (2024), finetuned directly from video diffusion models, fail to synthesize highly 3D consistent results **due to the limited receptive field of the temporal attention, which fails to provide sufficient information interaction during large camera movements.** Additionally, video diffusion models formulate the generation process as a sequence of video frames, which involves higher computational costs and greater uncertainty in the reconstruction process compared to sparse view synthesis and reconstruction methods.

In contrast, sparse novel view synthesis methods can synthesize highly consistent novel view images with arbitrary camera conditions using inflated 3D attention. However, the performance of such methods is often constrained by the **poor novel-view generalization ability provided by 2D diffusion models**, making it difficult to synthesize satisfying results on out-of-domain inputs Shi et al. (2023b); Long et al. (2024). This raises the question: **Can we unlock generative priors in video diffusion models to enhance the generation quality and stability of sparse novel view synthesis methods?**

To address this, we first conduct an in-depth investigation into the generation process of SV3D:

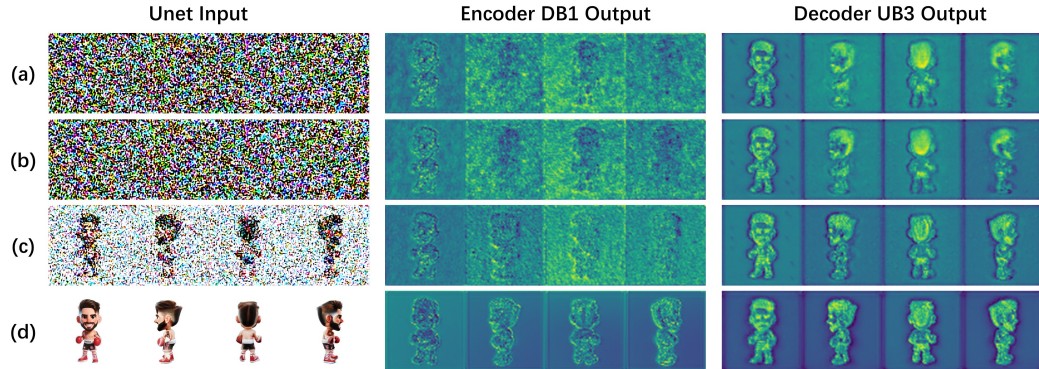

Figure 3: Visualization of feature maps in the generation process of the video diffusion model. (a-d) indicate the denoise steps of $50, 34, 18, 0$. From left to right, we show the input of the U-Net, the feature maps of the first downsample block in the encoder, and the feature maps of the third upsample block in the decoder of the U-Net. As shown in the right column, the feature maps from the temporal attention layer in the decoder block contain rich geometrical priors even with pure Gaussian noise as inputs (row a).

**Observation.** *Temporal layers in the decoder of the video diffusion U-Net are capable of providing rich geometrical priors for novel view synthesis even with noisy images as inputs.*

We provide empirical evidence to support this observation in Fig 3. We visualize the feature maps from different layers in the U-Net structure of the video diffusion model, SV3D. A surprising discovery is that the video diffusion model can synthesize rich geometrical structures even with pure Gaussian noise as inputs. This inspires us to use a pretrained video diffusion model directly as a geometrical reference network to enhance the generation process of the sparse novel view synthesis network.

However, integrating the geometrical feature priors from the video diffusion model into the sparse view generation network is non-trivial. The integration should not compromise the original sparse novel view synthesis model's ability and should be efficient for training and inference. To address this, we propose a simple and efficient residual temporal adapter module as a plug-in residual temporal attention layer to guide the overall generation process.

Specifically, given the image feature $I$ from the target sparse view synthesis network, we first reshape the feature map by merging the spatial dimensions into the batch axis. The reshaped feature map $I_t \in \mathbf{R}^{(b \times h \times w) \times f \times c}$ (where $f$ represents the number of generated views) and the extracted geometrical prior features $P \in \mathbf{R}^{(b \times h \times w) \times f_p \times c}$ (where $f_p$ represents the frame number generated by the video diffusion model) from the pretrained video diffusion model are then fed into the plug-in residual temporal attention layer to calculate the temporal residuals for each synthesized novel view image, which can be formulated as:

$$I_t^{new} = Softmax(\frac{Q_t K_P^T}{\sqrt{d}})V_P + I_t,$$

$$where \quad Q_t = I_t W_q, K_P = PW_k, V_P = PW_v.$$

(1)

The computational and memory costs of the residual temporal attention layer are quite low as it operates across views but separately for each spatial location. To further modulate the control strength for priors extracted from different denoise stages, we propose an adaptive control module to predict the control mask for the extracted video priors and adjust the control strength. The adaptive control module is implemented with two MLP layers.

Denote $n$ as the denoise step of the pretrained video diffusion model. The adaptive control module, together with the temporal residual attention layer, can be reformulated as

$$I_f^{new} = M(n, t) \times Softmax(\frac{Q_f K_P^T}{\sqrt{d}})V_P + Z_f,$$

(2)

where $t$ denotes the denoise time step of the sparse view synthesis network and $M$ denotes the mask prediction network.

In our experiments, we empirically select the feature maps output from the temporal attention layer in the decoder block of the video diffusion model as the geometrical priors, as they contain the most complete information from the model. Although a considerable amount of geometrical information can be extracted from the video diffusion model using pure Gaussian noise as inputs, we found that adopting a small number of denoise steps further enhances the fidelity of the extracted priors. Therefore, we design a shifted denoise schedule with eight steps in total for the video diffusion model to quickly capture the geometrical shape information from the input images and we utilize the temporal feature at the eighth denoise step as the geometrical priors to enhance the generation quality and generalization ability of the sparse novel view synthesis networks.

During training, only the parameters of the newly added temporal residual layers are trained, while the pretrained video diffusion model and the sparse view synthesis model remains frozen. Such paradigm is efficient and preserves the ability and accuracy of the original sparse view synthesis networks. The mask prediction module is zero-initialized, providing an identity mapping at the beginning of training for fast convergence. Compared to prior methods that require well-designed augmentation strategies on the ground truth input images to bridge the domain gap between the reference signals of the training and inference stages, we directly adopt noisy images and extracted video features as the reference inputs during training, which aligns well with the inference scenario. This leads to stronger generalization ability for various inputs.

With the proposed temporal residual module, the geometrical priors from the video diffusion model are effectively captured and integrated into the generation of the sparse novel view networks, enhancing generalization ability and leading to better generation quality with strong 3D consistency.

## 3.2 SPARSE VIEW SYNTHESIS FRAMEWORK

Besides the proposed geometrical reference network, we further study the influence of different temporal attention layers in video diffusion models.

**Observation.** *Although shallow temporal attention layers only interact with local information within adjacent frames, deep temporal attention layers can provide global structure information under large camera movements.*

We conduct experiments under the same settings as SV3D, which generates a 360-degree orbit video of the input images with 21 frames. As shown in the Fig 4, we first visualize the receptive fields of different temporal layers in the left column, indicating that the receptive field of shallow

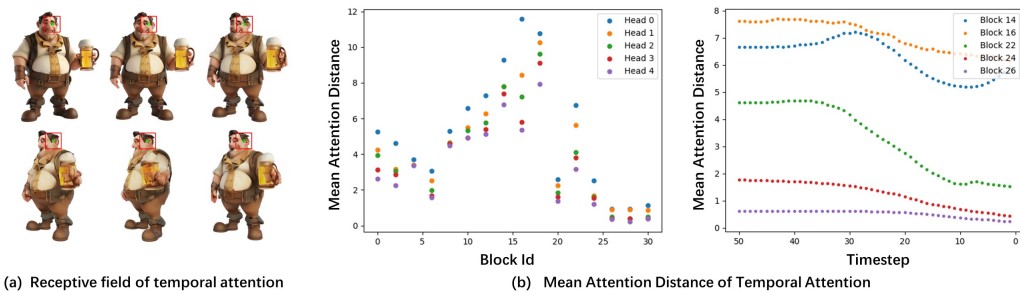

(a) Receptive field of temporal attention    (b) Mean Attention Distance of Temporal Attention

Figure 4: In the left column, we visualize the receptive field of temporal attention across different views, where the red box denotes the receptive field of the deepest temporal attention layer and the green box denotes the shallowest. In the right column, we show the mean attention distance (frames) of different temporal attention layers as well as in different denoise steps, where the Block ID follows the forward order in the video U-Net structure.

temporal layers only enhances consistency within adjacent frames. This observation demonstrates that in scenarios with large camera movements, shallow temporal attention fails to preserve 3D consistency in the generated results. We further analyze the mean attention distance of different temporal attention layers and identify that shallow temporal attention layers capture information within adjacent frames, while deeper layers capture global information with long attention distances across different views, providing global structure priors for the generated results.

Inspired by this, we propose a novel baseline model for sparse novel view synthesis, named the temporal enhanced sparse novel view synthesis network. Specifically, we finetune the pretrained video diffusion model to synthesize sparse novel view images by keeping the original temporal structure of the video diffusion model unchanged to preserve the global temporal knowledge in the pretrained deeper temporal attention layers. We extend the 2D spatial attention layer into a 3D attention layer by concatenating keys and values from different views to learn strong 3D consistency. This allows the network to benefit from the temporal knowledge initialized from the pretrained video model and generate more realistic images.

To support arbitrary trajectory generation, we replace the original fixed frame embedding with a camera-aware frame embedding conditioned on the target camera pose, similar to Zero123 Liu et al. (2023). This modification helps reduce temporal ambiguity caused by the fixed frame embedding for different camera views and allows the proposed sparse view generation network to synthesize novel views with arbitrary camera positions.

The overall framework of our proposed sparse view synthesis method, U3D, is shown in Fig 2. Specifically, we unify the proposed geometrical reference network with the new baseline model, the temporal enhanced sparse novel view synthesis network, into a novel sparse view synthesis framework named U3D. This framework unlocks the temporal priors from video diffusion models to generate high-fidelity multi-view images. With the proposed framework, we can generate highly consistent novel views from a single image that can be reconstructed into 3D assets via fast feed-forward sparse view reconstruction models. During our experiments, we adopt GRM Xu et al. (2024b) as sparse view reconstruction model to lift the generated multi-view images into 3D space.

## 4 EXPERIMENTS

### 4.1 IMPLEMENTATION DETAILS

We conduct training on the open-source multi-view dataset G-Objaverse Qiu et al. (2024), which is rendered from the ground truth 3D objects in Objaverse Deitke et al. (2023). We first reproduce SV3D as our base video diffusion model. Unlike SV3D, which directly inputs camera elevation and azimuth angles as conditions, our reproduced version adopts the pluckier ray embedding Tang et al. (2024) for camera control, which achieves similar performance. The temporal enhanced sparse view generation network is trained with 30k steps and a batch size of 128, serving as a baseline model

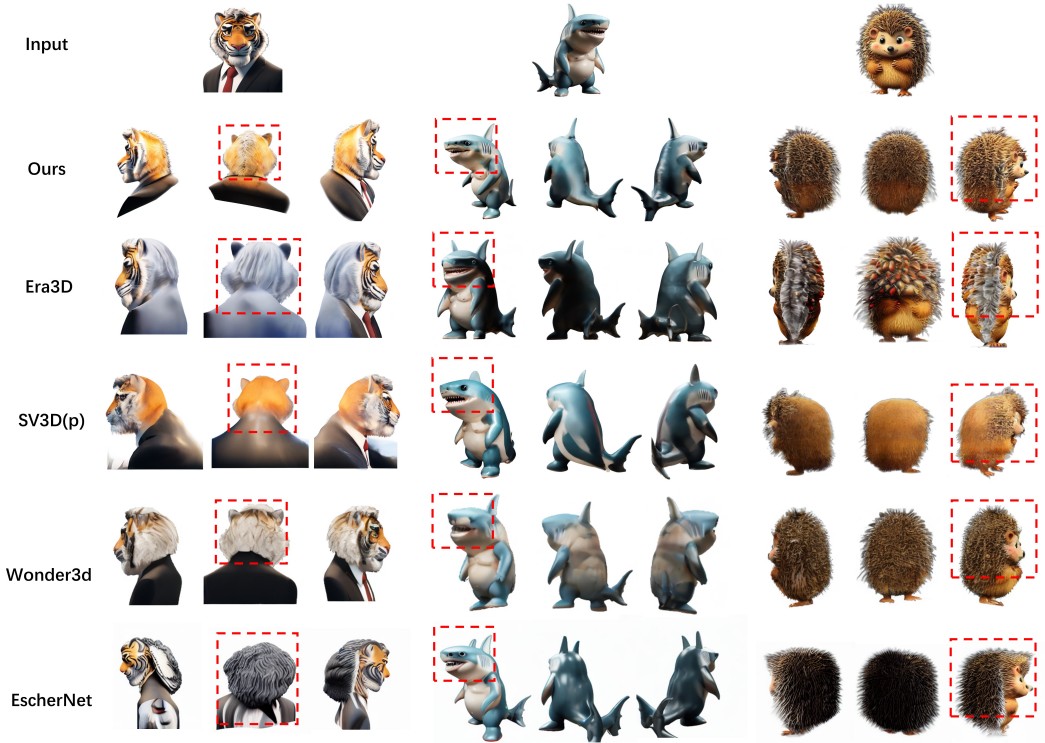

Figure 5: Qualitative comparisons of generated novel views between our models with State Of ArT novel view synthesis methods.

for training the residual temporal adapter. The training of the residual temporal adapter converges very fast with 6k steps and a batch size of 64. We utilize the AdamW optimizer and employ FP16 for efficient gradient descent without weight decay. The learning rate for all experiments is $1e-5$. Following Stable Video Diffusion, we adopt the EDM Karras et al. (2022) framework as the denoise sampling scheduler in both the training and inference stages.

### 4.2 QUALITATIVE COMPARISONS

We provide qualitative comparisons between our proposed U3D and other state-of-the-art novel view synthesis models, including EscherNet Kong et al. (2024), Wonder3d Long et al. (2024), SV3D(p) Voleti et al. (2024) and Era3d Li et al. (2024), as shown in Figure 5. Leveraging the strong generative priors from the large video diffusion model, U3D synthesizes high-quality novel view images with strong 3D consistency and better generalization abilities.

Constrained by the limited receptive fields of temporal attention layers, the video diffusion-based method SV3D(p) fails to capture 3D consistency with large camera movements and generates over-smooth results, as shown in Figure 5. On the other hand, limited by the poor generalization ability of 2D diffusion models, sparse novel view synthesis methods such as Era3d and Wonder3d fail to synthesize reasonable results on out-of-domain inputs, leading to collapsed structures and incorrect colors in the back view of the input image.

In contrast, benefiting from the proposed geometrical reference network and temporal enhanced sparse view synthesis network, U3D preserves the generative priors from the video diffusion model and generates high-quality novel view images with highly consistent 3D geometries and realistic colors. We further provide qualitative comparisons on the final reconstructed meshes. As shown in the Figure 6, our model demonstrates a great ability to generate 3D consistent novel view images, which can be reconstructed into high-quality meshes with correct geometric structures and are faithful to the input images.

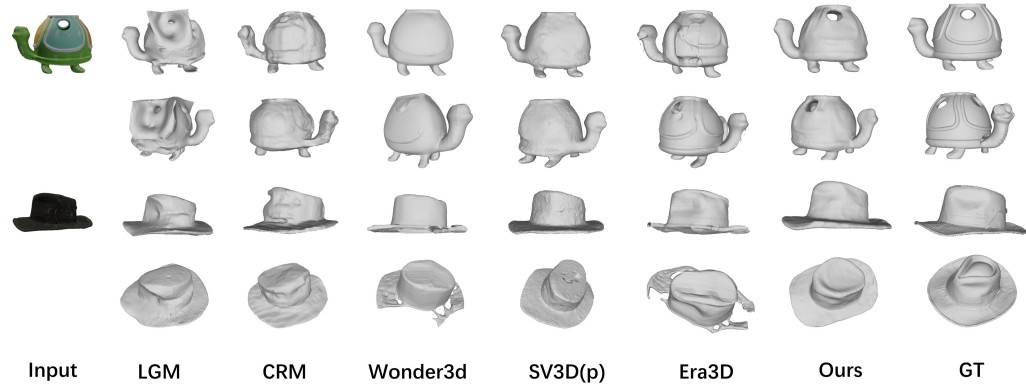

Input    LGM    CRM    Wonder3d    SV3D(p)    Era3d    Ours    GT

Figure 6: Qualitative comparisons of the generated meshes.

## 4.3 QUANTITATIVE COMPARISONS

We perform quantitative evaluation on the Google Scanned Objects dataset Downs et al. (2022). Specifically, we remove duplicated objects with the same shape and randomly select 200 objects for novel view synthesis evaluation and 50 objects for 3D reconstruction evaluation. For novel view synthesis, we calculate the Peak Signal-to-Noise Ratio (PSNR), Structural SIMilarity (SSIM), Learned Perceptual Image Patch Similarity (LPIPS), and CLIP similarity score (CLIP-S) to measure the generated quality and multi-view consistency at both pixel and semantic levels. For 3D reconstruction evaluation, we compute the Chamfer Distances (CD) and Volume IoU between ground-truth shapes and reconstructed shapes. As shown in Table 1 and Table 2, leveraging the strong generative priors of pretrained video diffusion models, our model outperforms other baselines across all metrics.

Table 1: Quantitative evaluation of novel view synthesis.

| Method | PSNR↑ | SSIM↑ | LPIPS↓ | CLIP(S)↑ |
|---|---|---|---|---|
| Zero123 | 15.01 | 0.8765 | 0.192 | 0.800 |
| Syncdreamer | 15.43 | 0.8592 | 0.183 | 0.802 |
| EscherNet | 15.69 | 0.8633 | 0.191 | 0.817 |
| Wonder3d | 19.65 | 0.8923 | 0.121 | 0.850 |
| SV3D(p) | 19.11 | 0.8901 | 0.122 | 0.864 |
| Era3d | 20.43 | 0.9081 | 0.116 | 0.859 |
| U3D(w/o TH,GR) | 19.80 | 0.8990 | 0.113 | 0.871 |
| U3D(w/o GR) | 20.23 | 0.9013 | 0.108 | 0.873 |
| U3D | **20.78** | **0.9103** | **0.104** | **0.882** |

Table 2: Quantitative results of 3d reconstructions.

| Method | CD↓ | IoU↑ |
|---|---|---|
| Shape-E | 0.0651 | 0.210 |
| One-2-3-45++ | 0.0516 | 0.359 |
| Syncdreamer | 0.0529 | 0.361 |
| EscherNet | 0.0513 | 0.382 |
| LGM | 0.0425 | 0.451 |
| CRM | 0.0411 | 0.465 |
| Wonder3d | 0.0382 | 0.468 |
| SV3D(p) | 0.0375 | 0.463 |
| Era3d | 0.0369 | 0.472 |
| U3D | **0.0362** | **0.479** |

## 4.4 ABLATION STUDIES

**Geometrical Reference Network.** As shown in Fig 7 and Tabel 1, we evaluate the effectiveness of the proposed geometrical reference network. Without it, the model fails to synthesize correct geometry under different camera conditions. In contrast, the geometrical reference network provides rich geometrical information from the video diffusion models, guiding the sparse novel view synthesis network to generate correct geometry with strong generalization abilities. We further evaluate the influence of the number of denoising steps adopted on the video diffusion models to extract the priors. As shown in Fig 7, $N = 0, 8, 20$ denotes the adoption of a denoise schedule with $N$ steps before prior extraction. Compared to priors directly extracted from pure Gaussian noise, better geometrical information is obtained after a small number of denoise steps (eight here). Further increasing the denoise steps leads to minor improvements in the generated results.

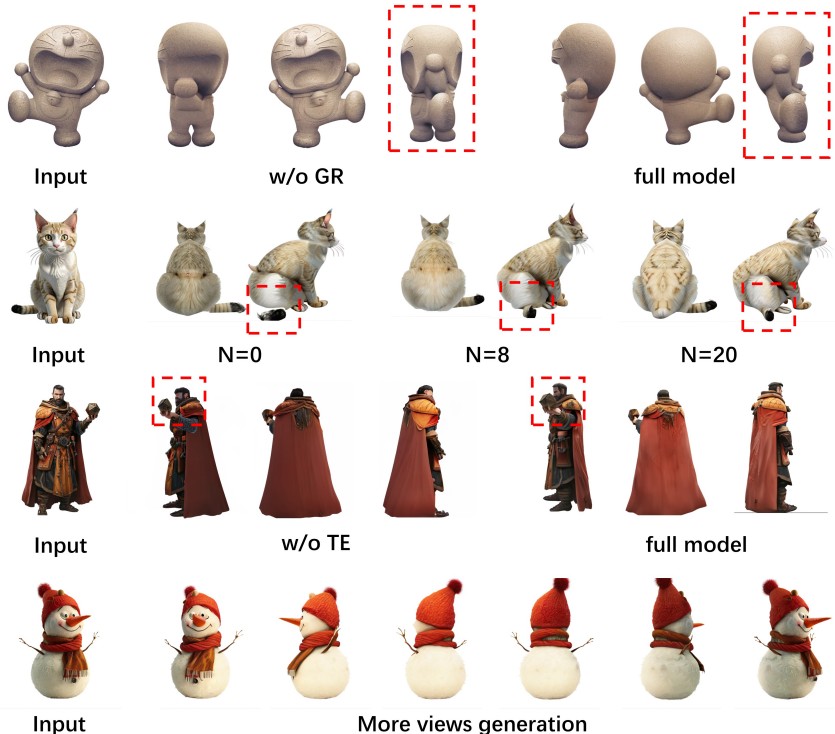

Figure 7: Ablation studies, where w/o GR represents without geometrical reference network and w/o TE represents without temporal enhanced sparse view synthesis network.

**Temporal Enhanced Sparse view Synthesis Network.** We compare the performance of the sparse view synthesis network when retaining or removing the temporal attention layers in the pretrained video diffusion model. As shown in the Fig 7 and Table 1, retaining the temporal attention layers results in better performance, synthesizing more realistic details and complex patterns. This demonstrates our observation that deep temporal attention layers provide generative priors that facilitate sparse view synthesis.

**Num of views.** Although we only adopt four views in the training process, the trained sparse view synthesis framework can be directly extended to generate more views with strong 3D consistency. In Fig 7, we show the results of generating six novel views conditioned on the input images.

## 5 LIMITATIONS

Although our model achieves promising results in sparse novel view synthesis, its performance is still limited by the quality of the video priors. A better video diffusion model may lead to improved results. Additionally, our model struggles to generate intricate structures, especially for thin objects. Enhancing the novel view synthesis network with 3D understanding capabilities may be a promising future research direction to address this issue.

## 6 CONCLUSION

In this paper, we present U3D, a novel sparse view synthesis method that unlocks generative priors from pretrained video diffusion models to enhance the generation of sparse novel views. The proposed U3D framework consists of a geometrical reference network and a temporally enhanced sparse novel view synthesis network. Leveraging the strong geometrical priors from the pretrained video diffusion model, U3D can generate highly consistent novel view images, which can be reconstructed with feed-forward sparse view reconstruction methods to produce high-quality 3D assets.

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

## A    APPENDIX

In this appendix, we provide more generation results, including more reconstruction results (Fig 8), more qualitative comparisons with video diffusion model (Fig 9 and Fig 10) and visualization of generated results together with the extracted geometrical priors (Fig 11 and Fig 12).

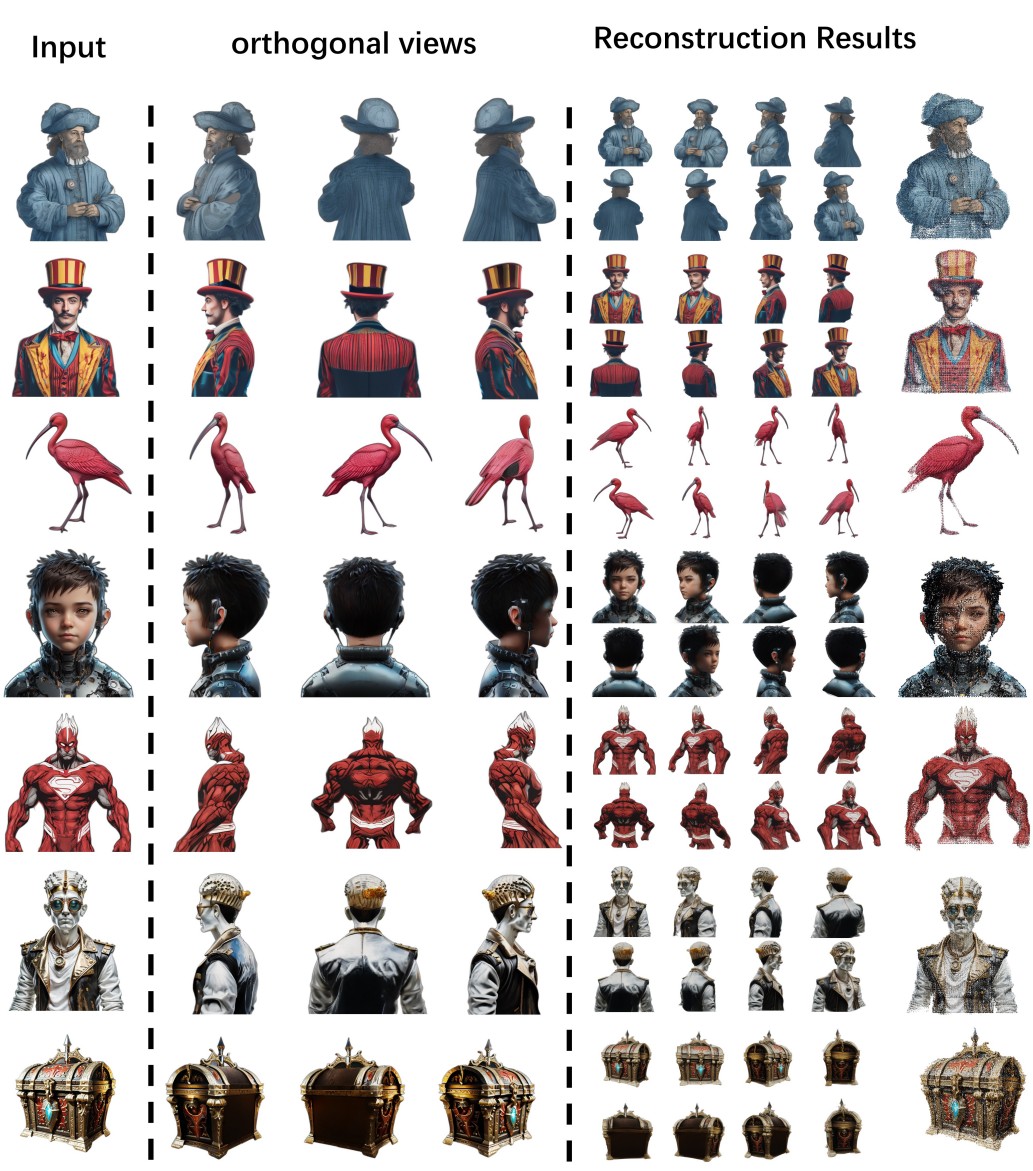

Figure 8: More orthogonal views generation and reconstruction results. (Please check the videos in supplemental materials for more reconstruction results.

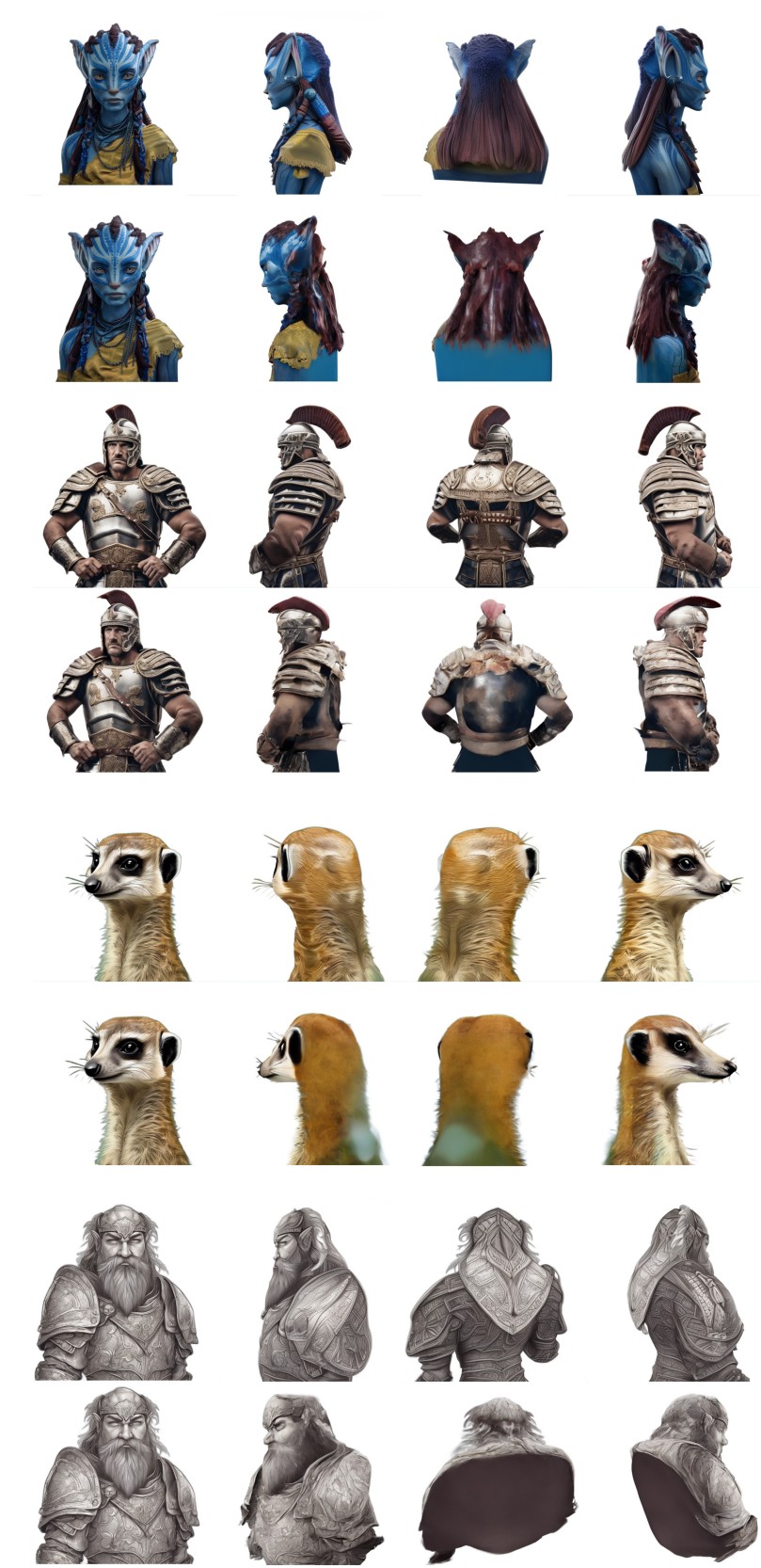

Figure 9: More qualitative comparison results with SV3D(p). The first column shows the input images. Every two rows show the generation results of four orthogonal views generated by our method and SV3D(p).

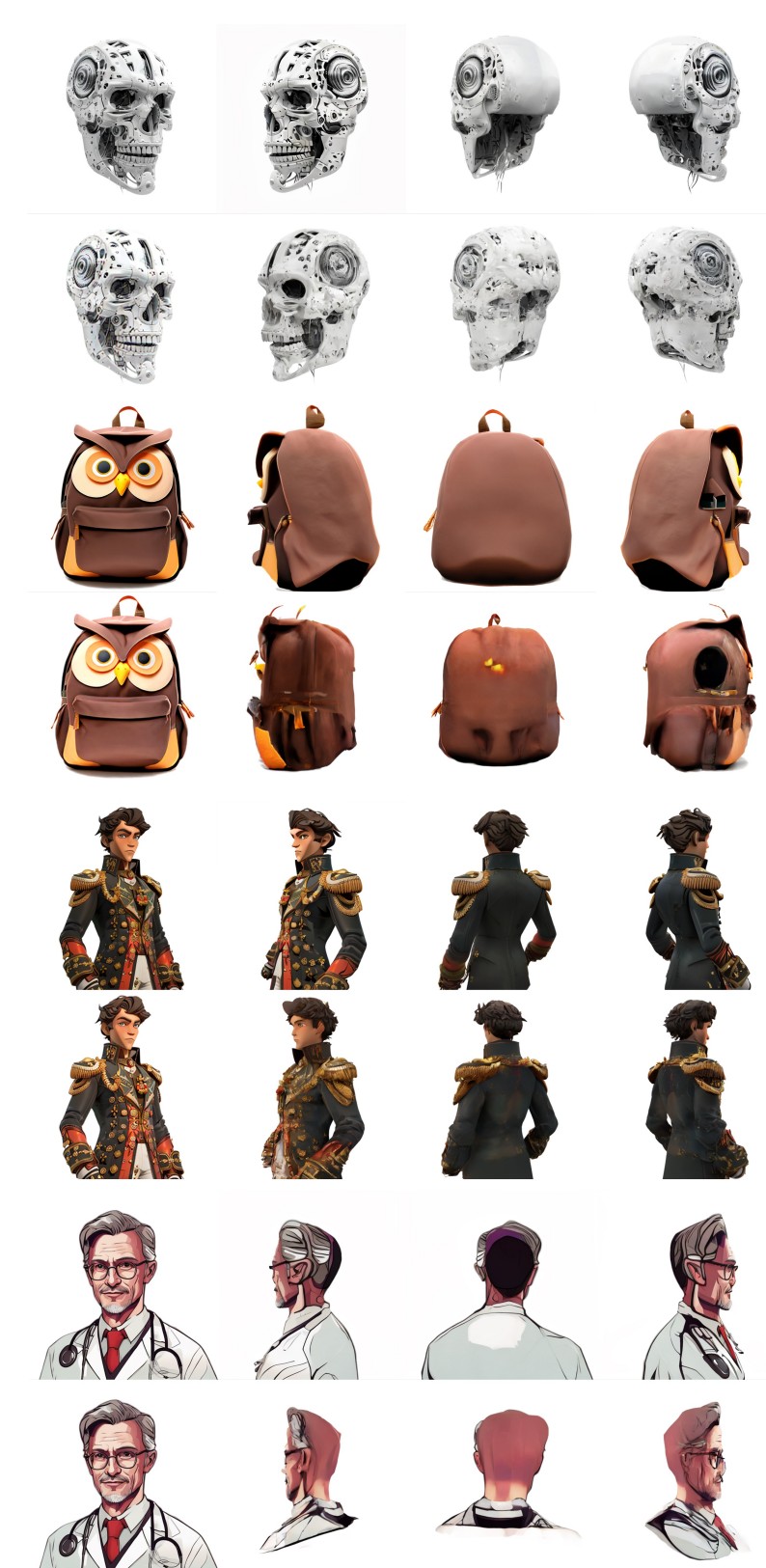

Figure 10: More qualitative comparison results with SV3D(p). The first column shows the input images. Every two rows show the generation results of four orthogonal views generated by our method and SV3D(p).

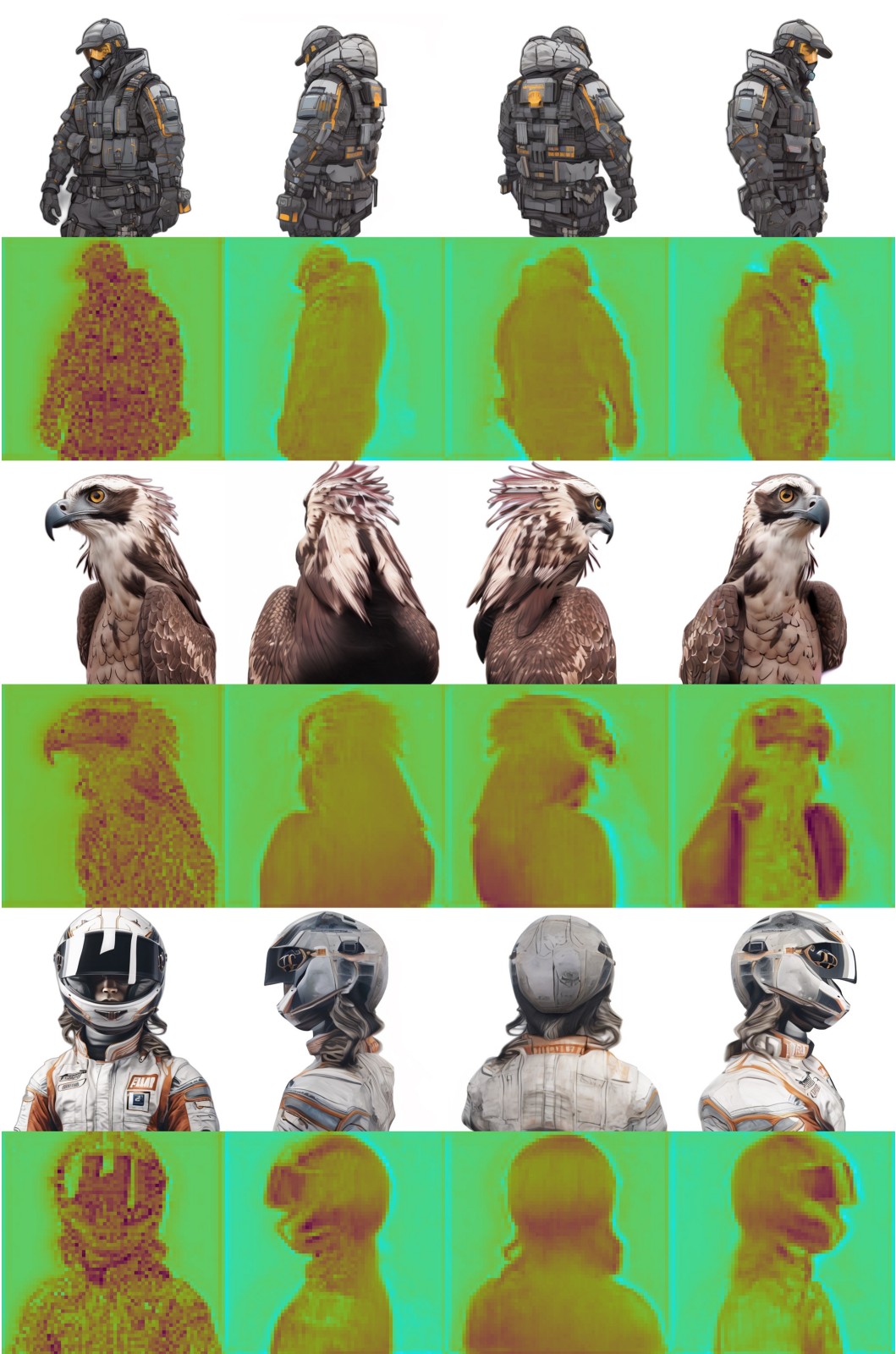

Figure 11: The orthogonal views generation results and the extracted video priors. Every two rows show the generation results of four orthogonal views and the corresponding geometrical video prior extracted from the pretrained video diffusion model.

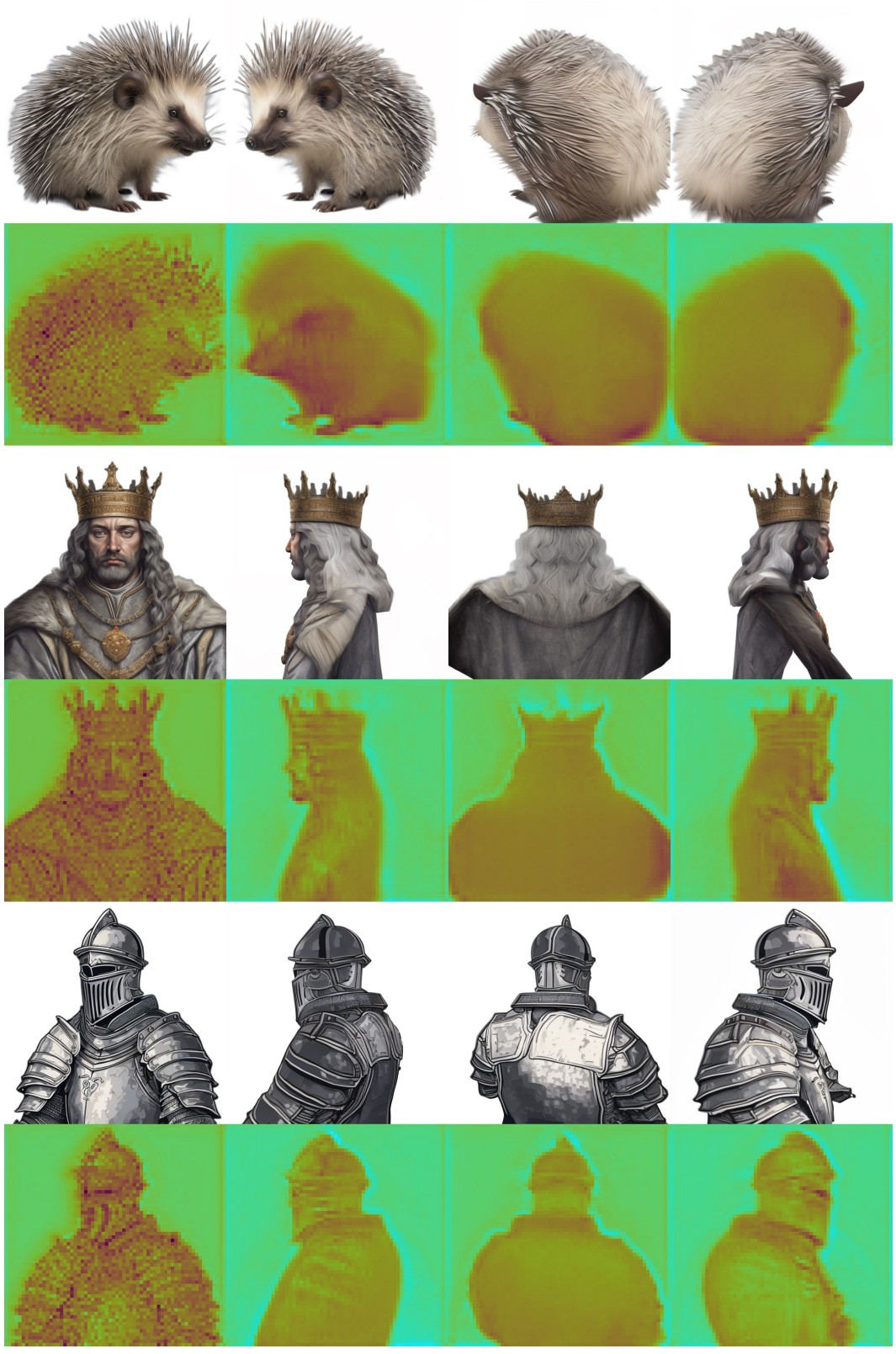

Figure 12: The orthogonal views generation results and the extracted video priors. Every two rows show the generation results of four orthogonal views and the corresponding geometrical video prior extracted from the pretrained video diffusion model.

