# OpenReview forum: "U3D: Unlocking the Video Prior for High Fidelity Sparse Novel View Synthesis and 3D Generation"
_ICLR.cc/2025/Conference — ICLR 2025 Conference Withdrawn Submission_

### Official Review · Reviewer_xngu · 2024-11-03

**Soundness:** 2
**Presentation:** 2
**Contribution:** 2
**Rating:** 5
**Confidence:** 5

**Summary:**

This paper proposes U3D, a framework to generate sparse but consistent novel views from a single image based on video diffusion models. Specifically, it uses a pre-trained video diffusion model (SV3D) as a geometrical reference network to provide geometrical priors (by denoising dense views for multiple steps) during training, and thus enhancing the view consistency of sparse novel view synthesis. The generated novel views are then used to reconstruct 3D GS via an off-the-shelf feedforward model (GRM). Both qualitative and quantitative results on the GSO dataset show slightly better performance of U3D compared to prior arts.

**Strengths:**

S1: Sensible model design
The idea of using pre-trained multi-view video diffusion model as reference is interesting and sensible. And the results show slight improvement over the baselines.

S2: Good writing
The paper is well-written and easy to follow overall.

**Weaknesses:**

W1: Training Efficiency
The proposed training strategy is not well-ablated, especially the choice of number of updates N=8. More importantly, the tradeoff between training efficiency and performance (controlled by N) is not reported in the paper. Since it requires N forward passes of the reference network (SV3D) and Dynamic control module during each training iteration, I suspect that the overall training is quite slow.

W2: Limited quantitative evaluation
I find the quantitative evaluation not sufficient to draw conclusion that U3D performs better than prior methods. First, it’s only evaluated on one dataset whereas SV3D also show results on OmniObject3D and real-world objects. Second, most metrics only indicate a marginal performance gain over the prior state-of-the-arts. Finally, it is unclear how the metrics in Table 1 are calculated. For instance, how many views are generated and how are they sampled? For SV3D, are the sparse views evenly sampled from the 21 generated views?

W3: Novelty in 3D generation
The main novelty of the paper lies in novel view synthesis and not 3D generation as it adopts an off-the-shelf reconstruction model (GRM). I would suggestion the authors to tone down the claim of contribution in 3D generation in the title and manuscript. Furthermore, it is crucial to also report 3D reconstruction metrics of baseline methods like SV3D by using its synthesized images plus the same reconstruction model for fair comparison.

**Questions:**

Q1: Following W1, what is the overall training time and how does the choice of N affect it as well as memory footprint during training?

Q2: Following W2, please clarify how the metrics in Table 1 are calculated.

---

### Official Review · Reviewer_EwSR · 2024-11-03

**Soundness:** 2
**Presentation:** 2
**Contribution:** 2
**Rating:** 5
**Confidence:** 4

**Summary:**

This paper addresses the problem of object-centric novel view synthesis from a single input image. This problem is inherently generative, yet requires the model to produce consistent results. The authors build upon Stable Video Diffusion and SV3D with two major designs. First, they utilize SV3D to extract geometrically consistent features and fuse these with a sparse-view synthesis network. For the view synthesis network, they retain full attention layers but replace the original positional encoding with Plücker ray embeddings. The authors fine-tune the geometry extraction module and sparse-view synthesis network on images rendered from the Objaverse dataset, comparing their approach to recent works and demonstrating improved view synthesis and geometry consistency on the Google Scanned Objects (GSO) dataset.

**Strengths:**

The approach of using video generative models to assist in inferring 3D structure and view synthesis is well-aligned with recent advances in the field.

It shows better quantitative results on GSO evaluation compared with recent methods.

The paper includes detailed visualizations to illustrates the method's underlying intuition, such as the receptive field analysis in Figure 4. which is interesting to see.

**Weaknesses:**

I1.  think the writing needs more improvements. Lot's of sentences in this paper is very wordy. One example: line 221: "Video diffusion models have been demonstrated to provide generative priors and serve as strong initialization models for finetuning novel view synthesis models." This could be simplified to"Video diffusion models provides rich prior for view synthesis tasks."

2. The evaluation is only done at GSO, it would be better to also add ABO.

3. The point of using temporal attention is save compute is very very weak.(line 280). Let's do a simple math here, for the attention module with two linear layers to_kqv and output_project. Suppose the feature dimension is D and number of tokens is N. Then the attention module with the linear layers has cost of 4 N D^2 (linear) + N^2 D (Attention). When doing 1D attention, the cost of the linear layer is still the same, and only the cost of attention layer is saved. What's the number of tokens and number of hidden layers? From the equation,  for significant savings, the number of tokens need to be over at least two times larger than the hidden dimension so that attention operation will cost most of the compute and 1D attention can increase efficiency non-trivially. Furthermore, this efficiency gain only applies to the geometry reference network, which may not be a dominant part of the overall computational load.

**Questions:**

1. Can you compare the amount of compute and trainable parameters in Geometric Reference Network and Sparse Novel View Synthesis Network?
2. I am not fully convicend with the effiectiveness of the Geometric Reference Network. If you removed it and reallocated the saved computation to train the View Synthesis model longer or with additional parameters, could the View Synthesis model achieve comparable results?
3. For the number if denoise steps in Geometric Reference Network, can you also provide a quantitative results?
4. A minor question regarding this paper, in general I don't think PSNR is a good metric, since novel view synthesis given one image is inherent generative, and PSNR is for deterministic task. Some paper compare FID, but I also don't know if it's a good metric, for me only user-study seems to be effective. But this is the common problem for this field, not only for this paper, so I left this as a minor comment.

---

### Official Review · Reviewer_xdKC · 2024-11-04

**Soundness:** 3
**Presentation:** 3
**Contribution:** 2
**Rating:** 5
**Confidence:** 5

**Summary:**

This paper proposes to utilize the pre-trained video diffusion model to boost the performance of sparse novel view generation models (i.e., GRM). It does that by training a new geometrical reference network to integrate the geometry prior into the video diffusion model into GRM.  Experiments are conducted on the Google Scanned Objects dataset to validate the effectiveness of the proposal.

**Strengths:**

S1. The paper is well-written and mostly clear.

S2. Using pre-trained video diffusion models to improve 3D generation is a promising research direction.

**Weaknesses:**

W1. Existing works, such as SV3D, IM3D[A], V3D[B], and Hi3D[C], have already utilized video diffusion models for 3D generation. The authors should discuss and compare these related works.

W2. The motivation for the proposed geometrical reference network is not clear. Why do we only need the geometry prior in the video diffusion model? Why not use the texture/appearance prior?

W3. The author has fine-tuned the video diffusion model with sparse views. This fine-tuned version seems to already be able to perform 3D generation. What is the performance of this model?

W4. There is no discussion about why choose GRM as the sparse view reconstruction model.

[A] IM-3D: Iterative Multiview Diffusion and Reconstruction for High-Quality 3D Generation.

[B] V3D: Video Diffusion Models are Effective 3D Generators.

[C] Hi3D: Pursuing High-Resolution Image-to-3D Generation with Video Diffusion Models.

**Questions:**

Please see the Weaknesses.

---

### Official Review · Reviewer_HH75 · 2024-11-04

**Soundness:** 3
**Presentation:** 2
**Contribution:** 3
**Rating:** 5
**Confidence:** 3

**Summary:**

The paper proposes a sparse novel view synthesis method utilizing pre-trained video diffusion models by incorporating a proposed residual temporal adapter, motivated by a series of empirical investigations on existing video diffusion models.

**Strengths:**

* The method is well-motivated with empirical investigation on SV3D via visualizations of feature maps and receptive fields.
* The empirical results show good performance improvement upon baselines.

**Weaknesses:**

* The paper claims that methods that finetune video diffusion models to generate orbit renderings lack 3D consistency, and the claim should be backed up by explanations with quantitative evaluation.
* The paper claims that integrating video diffusion models helps performance on out-of-distribution scenarios. Evaluation on inputs that are beyond Objaverse training distribution would help strengthen the argument. For example, does the method well handle styled images that are out of training distribution? This should be the case if the method does preserve the generalization capability of video diffusion models.

**Questions:**

* I assume the base video diffusion is Stable Video Diffusion to be consistent with SV3D. This can be more explicitly clarified.
* What's the mesh reconstruction pipeline? Is it the same for the proposed model and for baselines, e.g., SV3D? In the mesh reconstruction phase, how many views are used for the proposed method? How does the mesh quality change with different numbers of views?
* Notations in Eqn. (1-2) could be further clarified. For example, $W$ is currently not defined in Eqn. (1); $M, Z$ not defined in Eqn. (2).
* Supplementary results don't currently show comparisons with baselines. Animated comparisons on video outputs as well as reconstructed mesh would help with intuitive performance comparison.

---

### Note · Authors · 2024-11-13

I have read and agree with the venue's withdrawal policy on behalf of myself and my co-authors.